# Tailoring atomic layer growth at the liquid-metal interface

Hai Cao[1], Deepali Waghray[2], Stefan Knoppe [1,3], Wim Dehaen [2], Thierry Verbiest[1] & Steven De Feyter [1]

Engineering atomic structures at metal surfaces represents an important step in the development of novel nanomaterials and nanodevices, but relies predominantly on atomic/molecular beam epitaxy under ultrahigh vacuum conditions, where controlling the deposition processes remains challenging. By using solution-borne nanosized gold clusters as a precursor, here we develop a wet deposition protocol to the fabrication of atomically flat gold nanoislands, so as to utilize the dynamic exchange of surface-active molecules at the liquid-metal interface for manipulating the growth kinetics of ultrathin metallic nanostructures. While remarkable shape and size selection of gold nanoislands is observed, our experimental and theoretical investigations provide compelling evidences that organic adsorbates can impart a bias to the island orientation by preferred adsorption and alignment and intervene in the assembly and disassembly of adatom islands by complexing with Au adatoms. This approach offers a simple solution to regulate atomic layer growth of metals at ambient conditions.

[1] Division of Molecular Imaging and Photonics, Department of Chemistry, KU Leuven, Celestijnenlaan 200 F, B3001 Leuven, Belgium. [2] Division of Molecular Design and Synthesis, Department of Chemistry, KU Leuven, Celestijnenlaan 200 F, B3001 Leuven, Belgium. [3] Present address: Institute for Physical Chemistry, University of Stuttgart, Pfaffenwaldring 55, 70569 Stuttgart, Germany. Correspondence and requests for materials should be addressed to H.C. (email: caohai@iccas.ac.cn) or to S.D.F. (email: steven.defeyter@kuleuven.be)

When at least one dimension of a metallic structure approximates the de Broglie wavelength of electrons, energy states of the electrons become discrete and thickness-dependent[1,2]. In the last three decades, surface-supported nanostructures comprised of a limited number or a few layers of metal atoms, including two dimensional (2D) metallic islands and ultrathin films, have been the subject of intense study by means of scanning tunnelling microscopy (STM) and scanning tunnelling spectroscopy (STS)[3–7], not only for getting insight into the fundamentals of quantum mechanics[8–11], but also in quest for novel electronic[12,13], magnetic[14,15] and catalytic[16,17] properties to meet the demands of new and advanced materials. Varying the size and shape of such metallic nanostructures offers a feasible path to modulate their intrinsic properties[18–23], but controlling the growth of metals with dimensions in the order of some angstroms to some nanometres is very challenging.

To date, the most prevalent avenue to prepare ultrathin metallic films is via atomic/molecular beam epitaxy, which operates under ultrahigh vacuum (UHV) conditions at low temperature[24]. In contrast to crystal growth, epitaxial growth of metallic structures at the vacuum/metal interface is usually determined by kinetics, in part due to the barrier for adatoms to descend from the step edge and to migrate across the terrace and the step[3,24–26]. In addition, oriented growth is often arrested by subtle energy differences in adatom attachment to different types of facets[24]. Therefore complex metastable morphologies are usually formed, in particular at low temperatures. Organic modifiers which have been widely applied for surface restructuring and faceting[27–29] are promising to manipulate the growth kinetics of thin metal films, as the adsorption and organization of molecules can on the one hand lower the surface free energy therewith promote the wetting properties of thin films, and on the other hand skew the growth preference with directional metal-organic interactions. Nevertheless, the simultaneous vapor-phase deposition of two totally different substances onto the surface in a controlled fashion is not easy.

Here we develop a wet deposition protocol, by using solution-borne nanometer-size gold clusters as a source of gold adatoms, and present a simple approach for controlling the growth of gold nanoislands at surfaces, by applying helicenes containing thioether moieties as an additive to skew the atomic layer deposition at the liquid-metal interface (Fig. 1a–d). The gold clusters fall apart into gold adatoms upon exposure to the gold substrate, which further diffuse, nucleate and grow into irregular islands (Fig. 1b, c). Application of surface-active helicenes as a modifier results in remarkable control over the morphology (Fig. 1d–f). In contrast to the deposition in vacuum, the active components in this study can be premixed at any ratio prior to atomic layer formation, and the simplicity of operation at the liquid-metal interface enables us to observe details of the atomic layer growth that have been missing in previous static STM investigations. The dynamic processes of epitaxial film growth are monitored by in situ STM and corroborated by DFT modelling, whereby we link the effect of molecular species on oriented growth of nanoislands to the adsorbate-directed diffusion and attachment of adatoms at the steps.

## Results

### Atomic layer formation at the liquid-metal interface.
The thiolate-protected gold cluster used in this study, $Au_{38}(SCH_2CH_2Ph)_{24}$ (left-handed or right-handed) containing 38 gold atoms and 24 2-phenylethanethiol (2-PET) ligands[30], was prepared following a thermal etching reaction with excess thiol removed by filtration and extensive washing[31]. The cluster is intrinsically chiral (Fig. 2a), shows a very narrow size distribution (2.5 ± 0.2 nm)[32], and is considered stable in solution by the 2-PET ligand shell. The racemic mixture of left-handed and right-handed enantiomers of $Au_{38}(SCH_2CH_2Ph)_{24}$, herein after referred to as (rac)-$Au_{38}$, was dissolved in 1,2,4-trichlorobenzene (TCB) at various concentrations and deposited on clean Au(111) surface. STM topography images, recorded under constant current conditions, of the modified Au(111) surfaces are shown in Fig. 2c and Supplementary Fig. 1, revealing large islands and lower-lying patches of parallel-aligned strands on the Au(111) surface (typical morphology of a clean Au(111) surface is presented in Fig. 2b), yet none of these structures matches the dimension of intact (rac)-$Au_{38}$ clusters.

The striped structures fit well with the commensurate (7 × √3) superstructure of 2-PETs on Au(111) surface (Fig. 2d)[33], implying the detachment of thiolates from the clusters upon exposure to the gold substrate. It appears that the Au nanoclusters become unstable without the protection of 2-PET ligands and fall apart into gold adatoms, which further diffuse, nucleate and grow into the 2-PET-covered bright islands on the surface. It can be inferred from the identical organization of 2-PETs on the islands and on the Au(111) terraces surrounding them that the growth of these islands is an epitaxial process. A height of ~0.25 nm of these islands is close to the d-spacing of bulk Au (Fig. 2c). Despite the fact that quite often edges of these islands run along the equivalent [11$\bar{2}$] directions, the morphology of these islands is by and large not defined.

### Complexation of helicene with gold.
Recently a growing number of studies have shown the complexation between polar organic adsorbates and metal adatoms[34–37], and in few cases the adsorbed organic species were found even able to immobilize surface metal adatoms into small islands[38]. As the thioether moiety represents an effective metal-coordinating ligand for trapping Au lattice-gas atoms[39–41], we selected diazadithia[7]helicenes, (P)-1 and (M)-1 (Fig. 3a)[42], as organic additives to steer the growth of gold islands. Our selection of helicenes as an additive is also based on the extensive knowledge of the adsorption and organization of helicenes at metal surfaces[43]. In addition, the characteristic bright feature of helicenes in STM images may facilitate in unraveling the way they interact with the metal. As will turn out further on, the chirality aspect is not crucial. First, we evaluated the metal-coordinating ability of these helicenes by investigating their self-assembly behavior on the Au(111) surface. Dropcasting a 20 μM solution of (P)-1 in TCB on Au(111) at room temperature furnished the gold surface mainly with a linear structure composed of rows of helicene molecules running along the [11$\bar{2}$] direction (Fig. 3b and Supplementary Fig. 1), but with a small shift between adjacent rows and an evident gap every four rows.

The intermolecular distance in a helicene row amounts to ~1.4 nm, slightly larger than that in a 1D close-aligned helicene chain (1.05–1.25 nm)[43,44]. In addition, bright spots that are isolated from the characteristic features of helicenes can be identified within each row (Fig. 3d), which we tentatively ascribe to gold adatoms that are coordinated to the thioether groups of one helicene and the thiophene unit of another.

In view of the fact that the formation of metal-organic coordination chains is often temperature-dependent[45,46], a complementary experiment was performed at a lower deposition temperature (4 ℃), that is, a cold TCB solution of (P)-1 is deposited onto a cold Au(111) substrate for 10 min. Further STM measurements performed at room temperature revealed a seemingly similar linear structure but with a shorter intermolecular distance (~1.1 nm) in each row and an angle of about 12.0° between the helicene row and the [1$\bar{1}$0] axis of the Au substrate

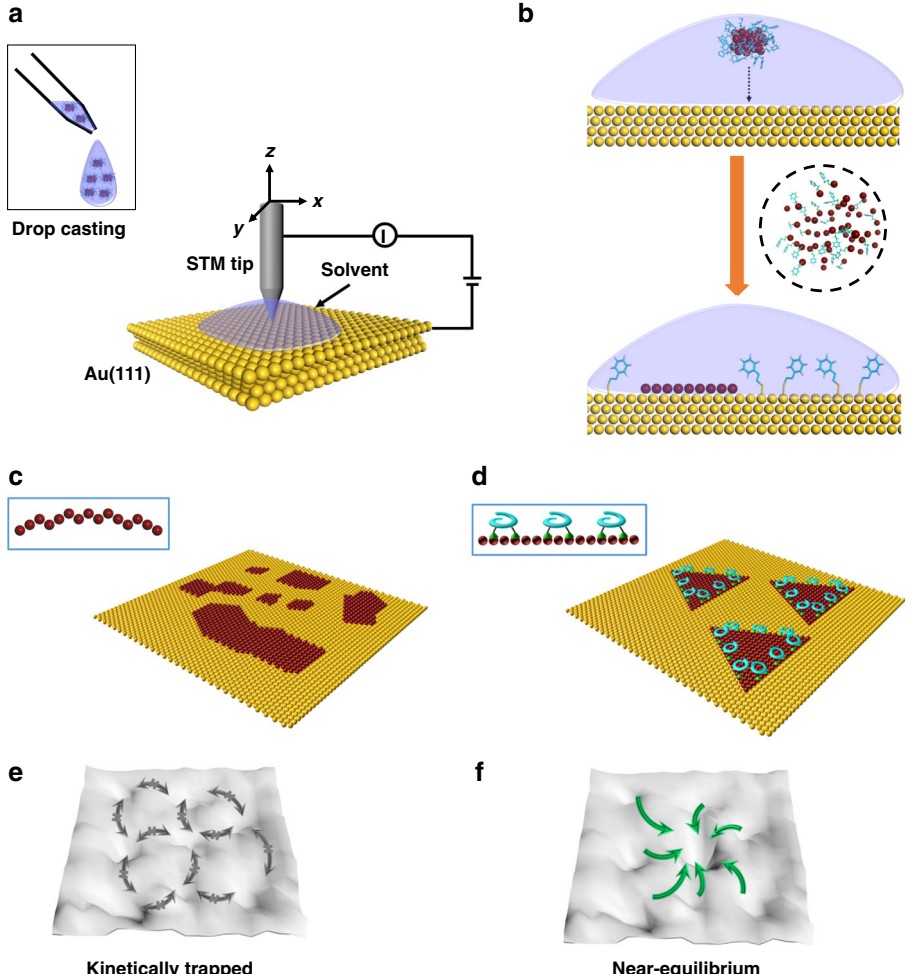

**Fig. 1** Atomic layer deposition at the liquid-metal interface. **a** Schematic illustration of STM measurement at the liquid-Au(111) interface. Solutions of different compounds are transferred to surface via dropcasting. **b** Growth of Au nanoislands via a wet deposition protocol. The atomic layer growth starts with the decomposition of thiolate-protected gold clusters at the solution-metal interface. Au atoms of the cluster and the island appear brown. **c**, **d** Growth of Au nanoislands in the absence (**c**) and presence (**d**) of surface-active helicenes (the blue 'helix' presentations) in solution. **e**, **f** Potential energy surfaces showing the growth kinetics of islands. The depth of the wells represents the relative stability of the islands, whereas the grey and green arrows indicate the pathways that are infeasible and feasible, respectively. Without organic modifiers, adatoms which are trapped at metastable binding sites cannot spontaneously switch to the optimum binding sites, giving rise to the formation of complex irregular metastable nanoislands. The adsorption and organization of helicenes at the step edges of the islands, on the other hand, impart a significant bias to the island growth along given directions, leading to the formation of triangular nanoislands

(Fig. 3c, f). On the basis of DFT calculations (see Supplementary Figs. 2, 3), structural models for these two linear structures of helicenes are provided in Fig. 3e, g, where the chemical interactions contributing to linear alignments of helicenes are metal coordination and complementary π-π stacking, respectively.

**Size and shape selection of adatom islands**. While the covalent gold-thiolate bonding has a strength close to that of the gold–gold metallic bonding, the gold-thioether bonding, in contrast, is a relatively weaker coordination-type interaction[40,47]. Therefore, ligand exchange in a solution containing (*rac*)-Au$_{38}$ and (*P*)-1 is not likely. Surprisingly, mixing of these two seemingly unrelated substances at different ratios in solution caused dramatic changes to the morphology of the islands. Figure 4a, b shows STM images of the surface morphology upon depositing an equimolar mixture of (*P*)-1 and (*rac*)-Au$_{38}$ in TCB at a total concentration of 50 µM onto a Au(111) surface at room temperature. Now, triangular islands are observed. The sides of the triangles appear to run

along the equivalent [1$\bar{1}$0] directions. Domains of 2-PET adlayer remain omnipresent on the regular Au(111) surface, but barely show up on the triangular islands or with a different appearance.

Close examination frequently reveals bright dots on the islands or around the islands but with different appearances: the molecules in the central area of an island often appear fuzzy, but the ones along the edges can be clearly identified. These bright dots can be removed by the STM tip (Fig. 4c) and appear with a regular spacing of ~1.4 nm (Fig. 4d). Though such a distance approximates the nearest neighbor distance in a metal-coordinated helicene row (~1.4 nm, Fig. 3d), the array of bright dots follows the equivalent [1$\bar{1}$0] rather than [11$\bar{2}$] directions. Taking into account the adsorption situation of helicene on pristine gold (see Supplementary Fig. 2), combined with the knowledge of the typical STM appearance of helicenes, the bright features at the steps are ascribed to helicenes that are side-by-side arranged (Fig. 4d).

The influence of helicene on the selective growth of adatom islands was confirmed in further experiments by varying the concentration of (*P*)-1 but keeping the concentration of (*rac*)-

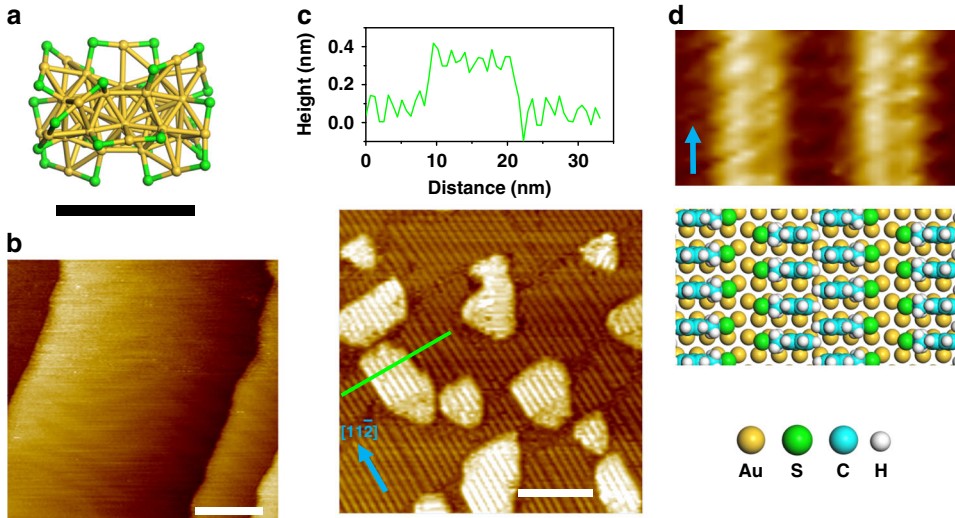

**Fig. 2** Atomic layer deposition at the liquid-metal interface. **a** Structure of the left-handed enantiomer of $Au_{38}(SCH_2CH_2Ph)_{24}$. For clarity, the phenylethyl groups are not displayed and a scale bar is used to illustrate the size of the cluster. **b** Representative STM image of a clean Au(111) surface. **c** STM image of the surface morphology upon the deposition of a 1,2,4-trichlorobenzene (TCB) solution of (rac)-$Au_{38}$ ($C = 5\,\mu M$) onto the Au(111) surface ($I_{set} = 130$ pA, $V_{bias} = -900$ mV). Height profile along the green line is displayed on the top. **d** Close-up STM image ($1.4 \times 2.5$ nm²) and structural model of the pin-stripe phase of 2-phenylethanethiol. Equivalent $[11\bar{2}]$ directions of the underlying Au(111) surface are indicated with blue arrows. Scale bars: **a**, 1 nm; **b**, **c**, 20 nm

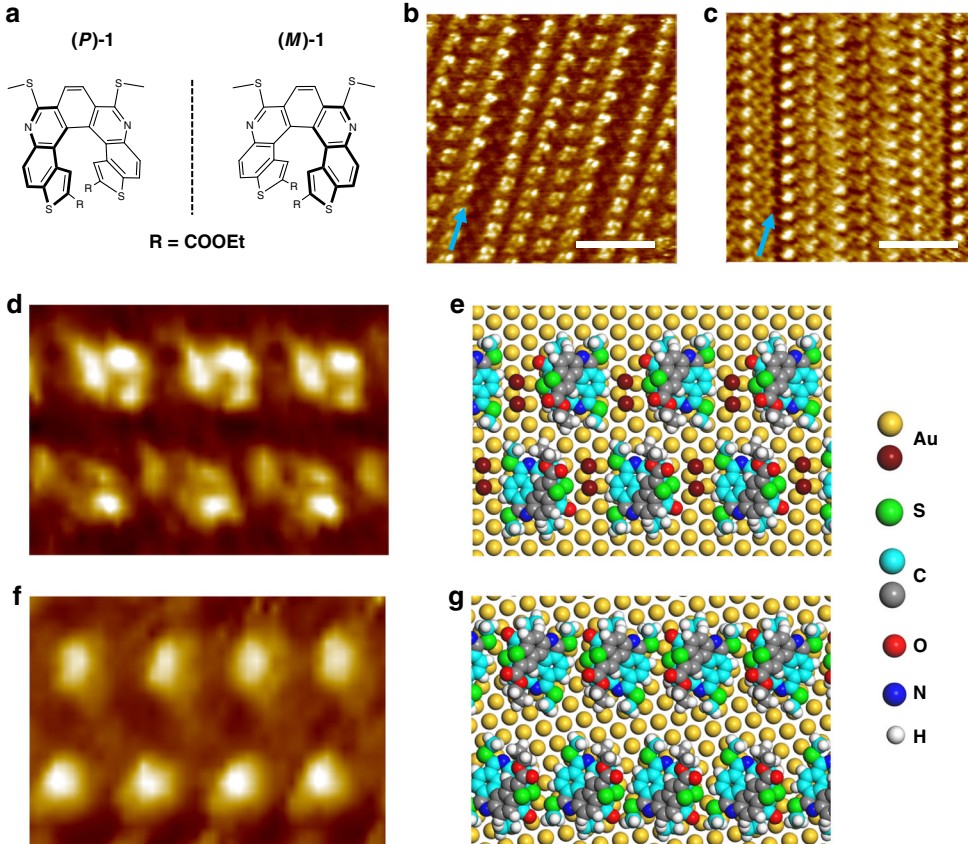

**Fig. 3** The helicene-Au complex. **a** Molecular structures of the two enantiomers of diazadithia[7]helicene, (P)-1 and (M)-1. **b**, **c** STM images of the linear superstructure of (P)-1 formed at room temperature ($C = 20\,\mu M$, $I_{set} = 100$ pA, $V_{bias} = -800$ mV) and at 4 °C ($C = 20\,\mu M$, $I_{set} = 100$ pA, $V_{bias} = -900$ mV) at the TCB/Au(111) interface, respectively. **d**–**g** Close-up STM image ($3.5 \times 5.0$ nm²) and structural model of the linear structure of (P)-1 formed in a room temperature solution (**d**, **e**) and at 4 °C (**f**, **g**), respectively. In the structural models, Au adatoms are dark brown. Carbon atoms are displayed in light blue, except for those of the protruding part of a helicene, which are displayed in grey. The brightest features in the STM images are attributed to the protruding part of the helicene molecules. Equivalent $[11\bar{2}]$ directions of the underlying Au(111) surface are indicated with blue arrows. Scale bars: **b**, **c**, 5 nm

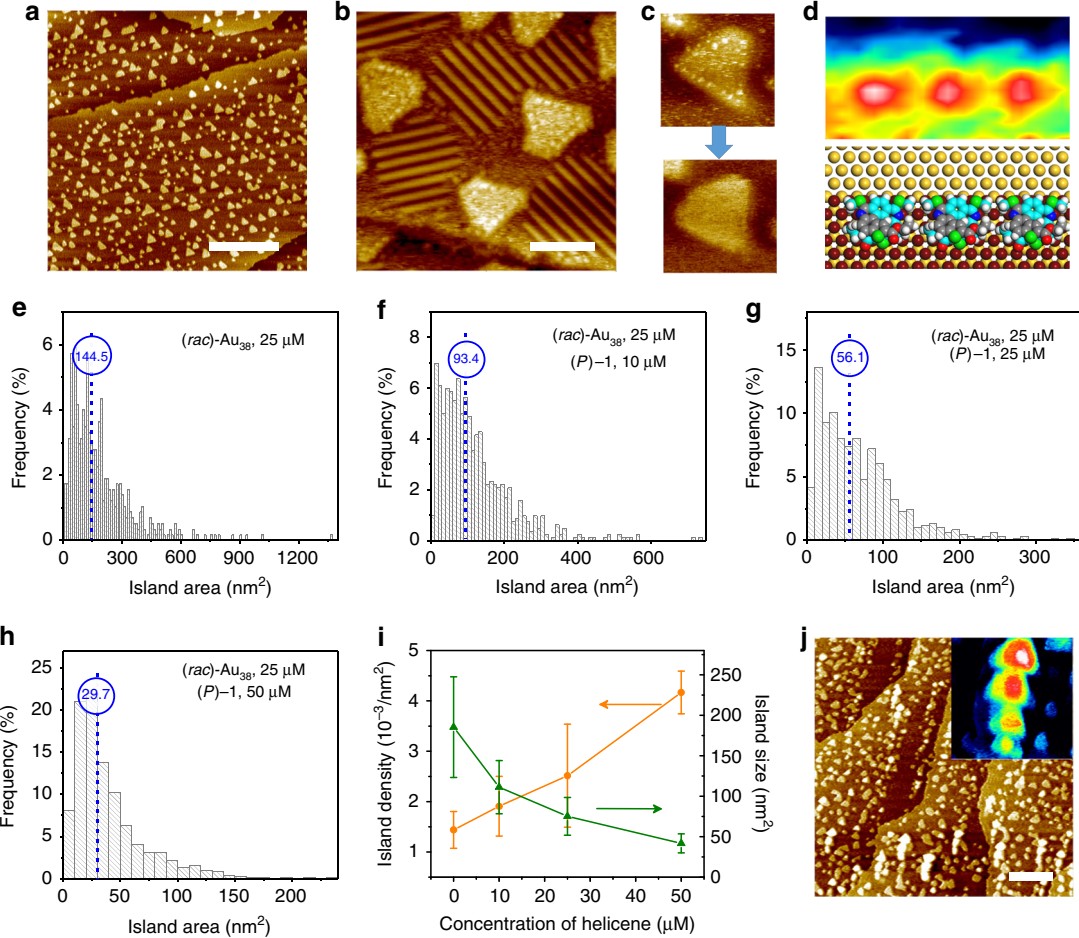

**Fig. 4** Morphological control: the shape, the size and the layers. **a**, **b** STM images acquired on Au(111) surface after the deposition of a premixed TCB solution containing 25 μM (*rac*)-Au$_{38}$ and 25 μM (*P*)-1. **a** $I_{set}$ = 100 pA, $V_{bias}$ = −1 mV; **b** $I_{set}$ = 100 pA, $V_{bias}$ = −900 mV. **c** STM images showing the islands before and after the removal, by the STM tip, of adsorbed species. **d** Contrast STM image (2.3 × 4.6 nm$^2$) and structural model of an array of (*P*)-1 at the step of a triangular island. **e–h** Histograms of the size of gold islands at various (*P*)-1 concentrations, from 0 μM (**e**), 10 μM (**f**), 25 μM (**g**) to 50 μM (**h**), while the concentration of (*rac*)-Au$_{38}$ is kept constant at 25 μM. The median sizes are indicated in blue. **i** Average island density and area versus the concentration of helicene in solution. The concentration of (*rac*)-Au$_{38}$ is 25 μM. The error bars were calculated as a s.d. of at least six images. **j** 3D mound formation in a mixture of 25 μM (*P*)-1 and 50 μM (*rac*)-Au$_{38}$. $I_{set}$ = 140 pA, $V_{bias}$ = −800 mV. The blue, yellow, red, and white colors in the inset outline four different layers. Scale bars: **a**, 100 nm; **b**, 10 nm; **j**, 50 nm

Au$_{38}$ constant (25 μM). It appears that sufficient helicene in the solution is necessary for efficient morphology control. The regular triangular morphology was only attained when the ratio of (*P*)-1 to (*rac*)-Au$_{38}$ is higher than 2:5 (see Supplementary Figs. 4–7). In addition to the shape control, the concentration of helicenes was found also to have a profound impact on the size of the islands, as shown by the area distribution analysis of the islands at varying helicene concentration (Fig. 4e–h). In the absence of helicenes, the size of the irregular islands spans a very wide range, varying from as low as 20 nm$^2$ to as high as 1300 nm$^2$. The area distribution is dramatically narrowed upon introducing helicenes and shifts to smaller values upon increasing the helicene concentration. For instance, 50 μM (*P*)-1 causes more than 70% of the islands to fall into the region of 5–45 nm$^2$, in stark contrast to the case without helicenes, where 55% of the islands have a size between 50–200 nm$^2$. While the average island size decreases in the presence of helicenes, the average island density increases almost linearly with the concentration of helicene additive (Fig. 4i and Supplementary Tables 1–4). All these findings seem to signal that the adsorption of helicene accounts for the selective growth of gold islands of particular shape and size. Next, we investigated the impact of (*rac*)-Au$_{38}$ concentration on island formation.

Increasing the concentration of (*rac*)-Au$_{38}$, from 10 μM (see Supplementary Fig. 6) to 25 μM (Fig. 4a) and then to 50 μM (Fig. 4j) in the presence of 25 μM helicene, resulted in a raise in surface coverage of the islands, from 15 ± 2% to 19 ± 4% to 29 ± 2% (see Supplementary Table 5 for details of the statistics). Remarkably, multiple-layer islands were formed before a full atomic layer coverage is reached. From the contrast STM image (Fig. 4j), up to four layers can be identified.

**The kinks**. To understand how the adsorbates affect the growth of islands, we inspected the steps where molecules are adsorbed. Considering that self-assembled monolayers of thiols have been intensively studied, we first examined the steps that are decorated with 2-PETs, for the purpose of comparison (Fig. 5a–c). The site preference for thiolate adsorption is the three-fold hollow one[27], and accordingly we constructed a model for a step along the [11$\bar{2}$] direction. A cut along the [11$\bar{2}$] direction inevitably results in kinks at the steps, which are stabilized by two rows of 2-PETs in the vicinity: one on the top of the island and one on the Au terrace plane.

Likewise, kinks in the middle of two rows of helicenes can be observed at the helicene-decorated steps, but the situation is more

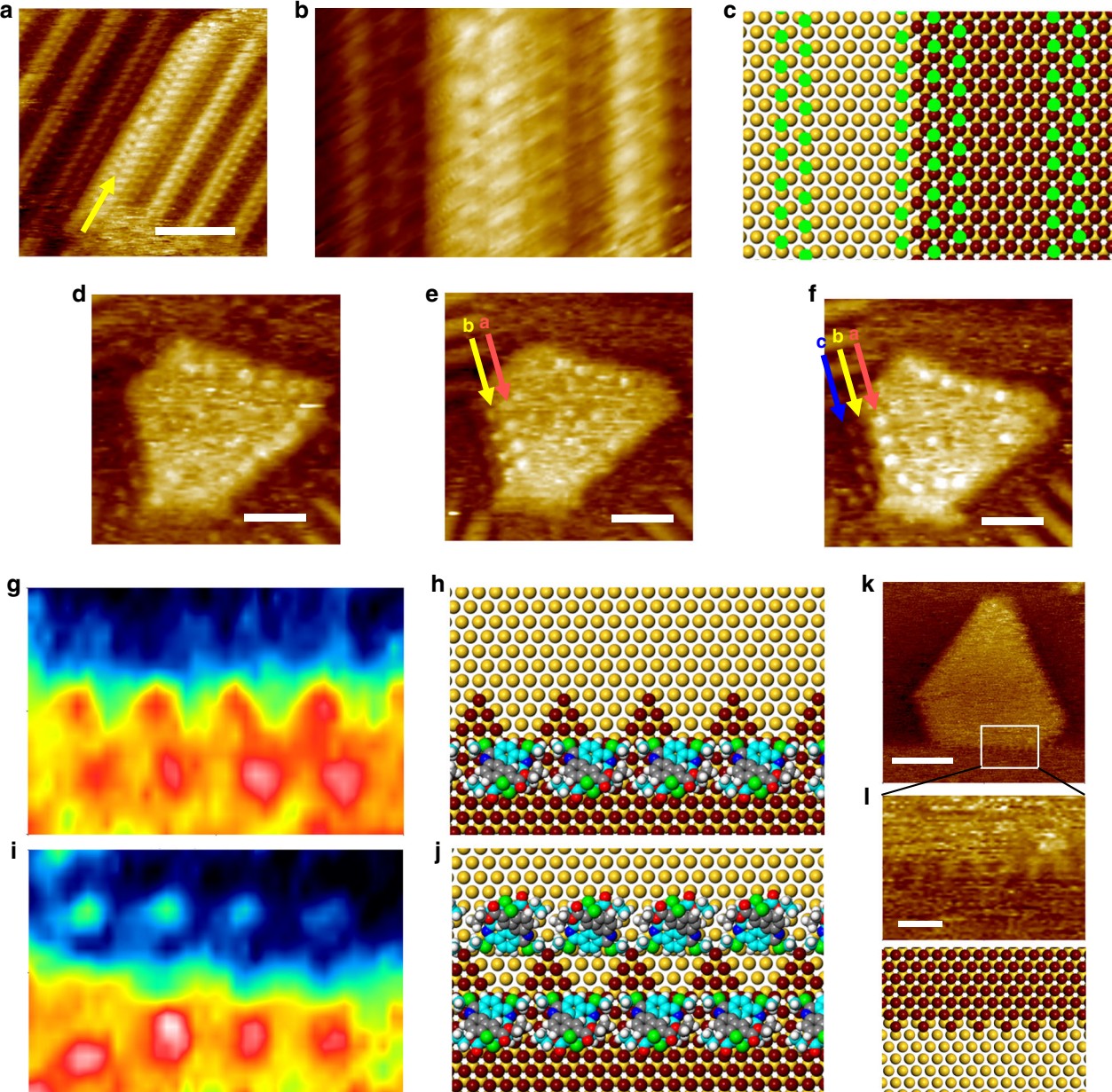

**Fig. 5** The kink sites. **a–c** STM images ($I_{set} = 140$ pA, $V_{bias} = -800$ mV) and illustration of the step edge decorated with 2-PETs. The yellow arrow points at the step edge, while the green disks indicate Au-S bonding sites. **d–f** Sequential STM images showing the changes at a step ($I_{set} = 100$ pA, $V_{bias} = -900$ mV) between a gold island and the gold surface underneath. Arrow a and c indicate helicenes while arrow b points to the kinks. Larger scale images of **d–f** are shown in Supplementary Fig. 8. **g, h** STM image ($4.0 \times 6.0$ nm$^2$) and structural model presenting sawtooth-like kinks at the edge that are decorated by a row of (*P*)-1. In the structural models, Au atoms of the island are marked dark brown. **i, j** STM image ($4.0 \times 6.0$ nm$^2$) and structural model showing an edge sustained by two rows of helicenes. **k** STM image of a triangular island after the complete removal of helicenes by the STM tip ($I_{set} = 250$ pA, $V_{bias} = -1$ V) and **l** close-up image and structural model of the new type of kinks. Scale bars: **a, d, e, f**, 4 nm; **k**, 5 nm; **l**, 1 nm

complicated, as the step is experiencing constant changes. Figure 5d–f is sequential STM images at the same location recorded in a time span of 4 min. With the adsorption of helicenes at the step, sawtooth-like kinks running along the [10$\bar{1}$] and [0$\bar{1}$1] directions appear in their vicinity (Fig. 5e, g and Supplementary Fig. 9). The dimension of these kinks matches 5 and $\frac{3\sqrt{3}}{2}$ primitive lattice vectors along the [1$\bar{1}$0] and the [11$\bar{2}$] directions, respectively. Based on the preferred configuration of this helicene on a Au(111) surface, a structural model is provided in Fig. 5h. Helicenes at the edge of an island are located at the base of the small triangular protrusions, i.e., the kinks, and are

arranged side-by-side. Further, one additional helicene row on the Au(111) terrace below can be identified from Fig. 4f acquired successively at the same location. These helicene molecules are presumably arranged in a way similar to those at the island edges (Fig. 5i, j). Clues for the impact of the helicenes on the island morphology can also be drawn from another observation: the appearance of smaller kinks at the step edge after the complete removal of adsorbed helicenes with the STM tip (Fig. 5k, l and Supplementary Fig. 10). The spacing between kinks approximates two primitive lattice vectors along the [1$\bar{1}$0] direction. Given that such kinks cannot form automatically, we believe they originate

from the degradation of the kinks that are induced and sustained by helicenes, i.e., the footprint of the helicenes still exists even after their desorption.

It is worth noting that even though chiral helicenes were used, the formation of kinks did not cause symmetry breaking at the steps, that is, the islands are achiral. The chiral moiety, namely the helical aromatic backbone, is located on the far side and has no influence on the kink formation. Therefore changes in the enantiopurity of the helicenes, from enantiopure to racemic, yielded no significant variation in the island formation (Supplementary Figs. 7, 11).

**DFT simulations**. The formation of kink sites has been generally recognized as a crucial step in crystal growth[3,48,49]. To unravel the correlation between helicene adsorption and kink formation at the steps, DFT calculations were performed to evaluate the impact of adsorbed helicenes on Au adatom attachment to the step. First, the adsorption geometry of helicene was examined. Epitaxial growth on the (111) surfaces of $fcc$ metals is known to be compounded by the competition between growth along two types of close-packed steps along the $[1\bar{1}0]$ directions: the (100)-faceted (A-type) and the (111)-faceted (B-type) (see Supplementary Fig. 12 for details)[24]. Triangular islands can either be A-type or B-type. Though (P)-1 interacts with the two types of steps in a similar fashion, the adsorption of (P)-1 at the B-step (Fig. 6a–c) was found 76 meV more stable than at the A-step (Supplementary Fig. 13). Such an energy difference, albeit small, is likely the reason that all triangles have the same orientation. The chemical interaction between a helicene molecule and the step was evaluated by the electron density difference analysis, where two Au-S bonds and a $\eta^2$-coordinated Au–C bond can be identified.

We then attempted to evaluate the influence of adsorbed helicene on adatom attachment to the step. The binding energy of a single Au atom to a B-step amounts to 3.658 eV. When it comes to the helicene-decorated step, six binding sites need to be considered, among which I–III are located on the side of the protruding part (or distal part) of helicene (hereinafter referred to as the distal side) while IV–VI are in the vicinity of the proximal part of helicene, or the proximal side. We found that the presence of helicene at the step barely affects the adatom attachment at position I, III, IV, and VI, but weakens the binding energy by 43 meV at II and 60 meV at V (Table 1). It seems that the attachment of Au adatoms to a step that is decorated with helicenes would weaken the chemical adsorption of helicenes. That is to say, the adsorption of helicene in the interior of an island should be weaker than that at the step, therewith the mobility of helicenes in the interior is higher. Such a subtle difference in adatom attachment, however, cannot be applied directly to explain the preference for kink formation at the distal side.

Nevertheless, using the lengths of two Au-S bonds as a reference, we noticed that the binding of a single Au atom to I, II, and III changes mainly the adsorption geometry of the distal part of a helicene, leaving the proximal part that is in direct contact with the substrate hardly affected. The binding of a Au atom to the proximal side, on the other hand changes almost only the part of helicene in the vicinity. Provided there is a Au atom attached to the helicene-decorated step already, the attachment of a second Au atom to the other side of the step was found dramatically weakened. For instance, the binding energy of a Au atom to V is weakened by 60 meV ($\Delta E_V$) in the presence of helicene but by 154 meV ($\Delta E_{V+II} - \Delta E_{II}$) when II is already occupied. There is a 94 meV ($\Delta E_{V+II} - \Delta E_{II} - \Delta E_V$) gap, which we ascribe to the repulsion for adatom attachment at different locations in the presence of adsorbed helicene. While the attachment of a Au

atom to one side strongly influences the binding of an Au atom on the other side, dimer formation on the same side, like at II and III and at IV and V are less affected. That is to say, the formation of dimer or longer strings at the step is more preferred on the same side, i.e., the distal side in this case. Such a bias for adatom attachment at the step, we believe, is the reason for preferred growth of kinks.

**Dynamics at the liquid-metal interface**. In view of the fact that the adsorbed helicene on the one hand changes the free energy of the island edge, and on the other hand contributes to the kink formation, it should be considered as a surfactant in island growth[7,50]. However, simply inducing and stabilizing kink sites does not suffice to control the size of islands. The solution-to-surface and on terrace diffusion of Au atoms stops when they are bound to surface Au atoms/nucleus, and the new aggregate may remain stable, grow larger or decay again. A surfactant may affect the island coalescence via increasing the on-terrace diffusion energy of adatoms[7]. The dependence of island density on the concentration of helicenes (Fig. 4i) suggests that the helicenes stabilize certain particle sizes to allow for a higher nucleation density. Evidence for the helicenes acting on island formation may lie in the dynamic effect of helicenes on small clusters at the liquid–solid interface. Figure 7a–d presents sequential STM images acquired after the deposition of a mixture of helicene and Au clusters onto the Au(111) surface. The images are recorded at the same location in a time span of 4 min, from which the adsorption and desorption of helicenes can be identified. The growth and decay of a small cluster appear to occur simultaneously with dynamic exchange of helicenes at the liquid-metal interface. However, the islands with relatively large sizes are hardly affected (see Supplementary Fig. 14). While it is likely that helicenes intervene both in the growth and decay of islands, the former process slows down when there are limited Au atoms/clusters in the solution phase. The latter process, on the other hand, should easily occur at sites with highly under-coordinated surface Au atoms. Nevertheless, there is a significant barrier for helicenes to pull out Au atoms that are already in an "equilibrium" or in a stable state, especially considering that the mobility of top layer Au atoms of Au(111) surface is lower compared to other facets (110, 100, etc.)[51]. In other words, the dynamic adsorption/desorption of helicenes is likely able to influence island growth at the initial phase of atomic layer deposition when there are plenty of Au atoms/clusters available in the solution and on the surface, but have no pronounced impact on the size distribution after stable islands are formed.

We note that sufficient mobility of the surfactant is necessary for the control of island size and shape. Take 2-PETs as a counterexample, its diffusion on the islands is arrested by the strong gold-thiolate chemical bonding, which creates a bias towards the growth along a given orientation and results in the formation of irregular islands with a wide size distribution (see Supplementary Fig. 15).

In conclusion, we have developed a distinct approach to control the epitaxial growth of Au adatom islands at the liquid-metal interface by using nanosized gold clusters as a precursor for gold atoms and surface-active helicenes as an organic modifier to direct the growth of the atomic layer. This study provides an empirical and theoretical foundation for explaining the role of helicenes in size and shape selection of Au adatom islands: intervening in the assembly and disassembly of adatom islands at the initial stages of island growth and imparting a bias to the island orientation by preferred adsorption and alignment. Despite a reduction of surface free energy at the step edge, we argue that the adsorbed molecular species do not necessary

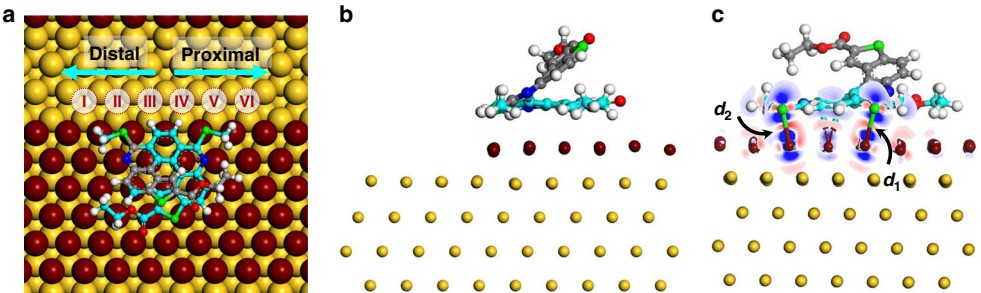

**Fig. 6** Chemical bonding at the step. **a** Top, **b** side and **c** front views of the adsorption geometry of a helicene at the B-step. I–VI are the different locations considered for the evaluation of adatom attachment. Electron-density difference is applied in **c** to illustrate the chemical bonding between helicene and step. Charge depletion and charge accumulation are displayed in blue and in red with a contour value of 0.03 e Å$^{-3}$, respectively. The two Au-S bonds are indicated by $d_1$ (2.728 Å) and $d_2$ (2.575 Å), respectively

---

### Table 1 Helicene-induced changes to the attachment of Au atom(s) to different sites of the step

|       | $\Delta d_1$ (Å) | $\Delta d_2$ (Å) | $\Delta E_n$ (meV) | $\Delta E_{n+m}$ (meV) | $\Delta E_{n+m} - \Delta E_m - \Delta E_n$ (meV) |
|-------|------------------|------------------|--------------------|------------------------|---------------------------------------------------|
| I     | 0.010            | 0.001            | −7                 | –                      | –                                                 |
| II    | 0.017            | 0                | −43                | –                      | –                                                 |
| III   | −0.056           | −0.001           | 15                 | –                      | –                                                 |
| IV    | −0.013           | 0.008            | −2                 | –                      | –                                                 |
| V     | −0.003           | 0.121            | −60                | –                      | –                                                 |
| VI    | −0.002           | 0.009            | −1                 | –                      | –                                                 |
| II + III | −0.047        | −0.003           | –                  | −40                    | −12                                               |
| IV + V   | −0.011        | 0.096            | –                  | −95                    | −33                                               |
| III + IV | 0.008         | 0.049            | –                  | −46                    | −59                                               |
| II + V   | 0.064         | 0.153            | –                  | −197                   | −94                                               |

$n, m = $ I, II, III...VI, $n \neq m$
$\Delta d_1 = d_1' - d_1$; $\Delta d_2 = d_2' - d_2$; $E_{bind} = -(E_{NAu \cdot step} - E_{NAu} - E_{step})$; $\Delta E_n = E_{bind}' - E_{bind}$
$E_{NAu}$, $E_{step}$ and $E_{NAu \cdot step}$ refer to the energy of $N$ ($N = 1, 2$) Au atom(s), the step and the step that is decorated with $N$ Au atom(s). $d_1'$, $d_2'$ and $E_{bind}'$ are the calculated results in the presence of adsorbed helicene

---

prohibit the Au adatom attachment in their vicinity, but instead drive the binding of Au adatoms in a way that affects the bonding situation of the adsorbate the least. While strong chemical interactions with the metal and sufficient on-surface mobility are both requisites for a molecular modifier, here we highlight the promising prospect of dynamic molecular exchange at the liquid-metal interface on manipulating the growth kinetics of metal nanostructures. The present findings may contribute to the field's understanding of the ways that molecular adsorbates act on the atomistic processes of atomic layer formation of metals, and may serve as a basis for additional research on engineering metal structures at surfaces.

## Methods

**Synthesis of (P)-1 and (M)-1**. Full details regarding the synthesis and characterization of helicenes are given in Supplementary Fig. 16 and Supplementary Note 1.

**Synthesis of racemic Au$_{38}$(SCH$_2$CH$_2$Ph)$_{24}$ clusters**. The gold clusters were prepared and isolated according to a published protocol[32]. The crude clusters were purified with size exclusion chromatography using tetrahydrofuran as mobile phase and Biorad S-X1 beads. The fraction containing Au$_{38}$(SCH$_2$CH$_2$Ph)$_{24}$ clusters was collected and purified in repeated SEC cycles until the beginning and the end of the band showed identical absorption spectra. The synthetic batch used in this study is the same that was used in ref. [52], see there for absorption spectrum and mass spectrometric analysis.

**Solution preparation**. 1,2,4-trichlorobenzene (TCB, Sigma-Aldrich, 99%, used as received) was used as solvent to dissolve all the compounds at various concentrations. The concentrations used in each experiment are specified in the figure

captions. The evaporation of TCB (b.p. 214 °C) is very slow at RT, and thus the loss of solvent that occurs during the STM experiment can be neglected.

**STM measurements**. All STM experiments were performed at room temperature (20–23 °C) using a PicoSPM (Molecular imaging, now Keysight) machine operating in constant-current mode with the tip immersed in the supernatant liquid. STM tips were prepared by mechanical cutting from Pt/Ir wire (80%/20%, diameter 0.2 mm). Imaging parameters ($I_{set}$: the setpoint tunneling current; $V_{bias}$: the bias voltage applied between sample and tip) are indicated in the figure captions. $V_{bias}$ is referenced to the sample. STM imaging is performed in the quasi-constant-current mode (height mode). Prior to imaging, a drop of the solution (~10 μL) was applied onto a flame-annealed Au(111) film on mica (~1 × 1 cm$^2$, Georg Albert PVD Company) for 30 min. It takes about 1 min to record a STM image. An STM session normally takes a day to obtain enough images, and both large scale images and close-up images were obtained at different sites, by moving the sample plate (after every six images) by a few millimeters.

For the cold deposition experiment, the helicene solution, the flame-annealed Au(111) substrate and the glass pipette were placed in a refrigerator at 4 °C for a few hours before the deposition. After the deposition of a drop of helicene solution onto the Au(111) substrate placed in a Petri dish for 10 min, the sample was transferred to room temperature for STM measurements. No apparent transition between both linear phases was observed during 4 h of STM imaging.

**Surface coverage analysis**. Surface coverage and size distribution of islands on surface were calculated by the SPIP software (Image Metrology, Denmark) with the 'particle & pore analysis' function. It was done in two steps. At the first step, the islands were detected by the software automatically. Then noise was removed and if necessary some islands were mapped out manually. The second step is particularly important when step edges are present. The area of each island was calculated by the software and used for further analysis. An example is shown in Supplementary Fig. 17.

**Details of DFT calculations**. If not otherwise specified, DFT calculations are performed using the Castep module available in Material Studio 2017 in a

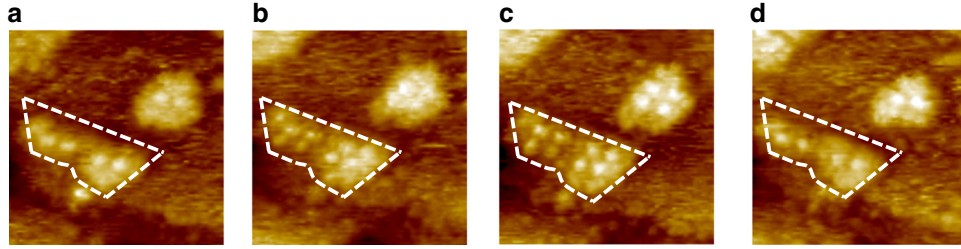

**Fig. 7** Dynamics. **a–d** Successive STM images ($15 \times 15$ nm$^2$, $I_{set} = 100$ pA, $V_{bias} = -900$ mV) at the same location showing the growth and decay of an adatom island, along with the adsorption and desorption of helicenes

generalized gradient approximation (GGA). The gradient-corrected Perdew Burke Enzerhof (PBE) exchange-correlation functional is employed together with an energy cutoff of 400 eV and Gamma k-points. The van de Waals interactions are described by a semi-empirical Grimme DFT-D2 approach[53–56]. The ionic core is represented by ultrasoft pseudopotentials. All the results are based on non-spin polarization calculations with density mixing scheme.

First, the performance of different methods and parameters were tested on calculating the bulk *fcc* lattice constant of Au (see Supplementary Tables 6 and 7), whereby we chose the C6 coefficient and the R0 van der Waals radius of 422 eV Å6 and 1.772 Å for Au, respectively. Lattice parameters **a** and **b** were determined by above calculations as 2.910 Å, corresponding to a bulk *fcc* lattice constant of 4.115 Å. The surface-supported island structure is described by a $7 \times 9$ supercell consisting of four monolayers of (111)-oriented Au slab, a $7 \times 6$ adatom layer and a vacuum layer of 20 Å. The bottom three Au layers are frozen in calculations while the surface layer, the Au island and the helicene are fully relaxed until the convergence criteria reach $2 \times 10^{-6}$ eV for SCF, $2 \times 10^{-5}$ eV/atom for electronic structure, and 0.05 eV/Å for forces. Isolated helicene molecule in the gas phase is optimized in a cell with dimensions (Å) $25 \times 25 \times 25$ and the Gamma k-point set only.

## Data availability

All relevant data that support the findings of this study are available from the corresponding authors upon request.

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

## Acknowledgements

We are grateful to the Fund of Scientific Research Flanders (FWO), KU Leuven - Internal Funds, the European Research Council under the European Union's Seventh Framework Programme (FP7/2007-2013)/ERC Grant Agreement No. 340324, the European Union Framework Programme for Research and Innovation Horizon 2020 as a Future and Emerging Technologies Action (FET Open) under Grant Agreement No. 664878 (2D-INK). This work was in part supported by FWO under EOS 30489208.

## Author contributions

H.C. and S.D.F. conceived and designed the concepts. D.W. and W.D. synthesized the helicenes. S.K. and T.V. prepared the gold clusters. H.C. acquired and analysed the STM data and performed the theoretical simulations. H.C. and S.D.F. co-wrote the paper with contributions from all the authors.
