## [Peer Review File · Nature Communications]

Reviewers' comments:

Reviewer #1 (Remarks to the Author):

The manuscript of Cao et al. reports on the growth of Au islands at the liquid-metal interface. The main result is the size distribution and shape of the islands grown in presence/absence of a molecular additive. The molecular additive is chiral, but its chirality is found to have no visible effect on the shape/faceting of the islands. A dependence on the helicity of the additive would have been a very interesting result. In the present state of the manuscript one wonders why such complex molecules have been chosen, it appears that many other additives would also yield the same effect.

The effect is interesting, but the paper raises a number of questions which should be addressed (see below). In the present state of the manuscript I'm not convinced if the size distribution is really related to the epitaxy. Here, the stabilization of smaller islands and 3D mound formation (at high concentrations) appears to be related to the stabilization of certain edges/facets (see 1). In that sense the work is similar to the cited literature on 'faceting'. On the other hand the deposition via Au clusters appears to be novel.

1) Is this effect reversible/dynamic, i.e. does the size distribution change if the helicenes are added after the islands have been grown? This experiment will tell if one can consider the growth here as nucleation and growth or if it's an equilibrium process depending on the concentration of helicenes. Figure 6 hints towards an equilibrium process.

2) If the growth is not dynamic, in vacuum based epitaxy the impingement rate of atoms to the surface vs. the diffusion at the surface have been considered. [1] In this picture, the 'nucleation density' is an important quantity. It could be that the presence of helicenes stabilizes certain cluster sizes and this allows for a higher nucleation density. This can be analyzed from the existing data.

3) Related to 2) also the dependence on the metal source (Au cluster concentration) which should probably regulate the rate at which the Au arrives is not discussed. Or is the rate at which Au arrives at the surface defined by the evaporation rate of the solvent? The methodical part of the paper is not very clear in my opinion.

4) Can the effect of the helicenes be discussed in terms a "surfactant"? The presence of helicenes clearly modifies the step/surface energies.

[1] Harald Brune and Klaus Kern; Chapter 5 Heteroepitaxial metal growth: the effects of strain; The Chemical Physics of Solid Surfaces 8; 149 (1997).

Other comments:

1) The figures are not reader-friendly: e.g. in Fig. 3d the reader has to extract the meaning of the curves from the caption. It would be much easier to give this information in the legend. The other figures can be also improved by adding some text to the figures themselves.

2) In Figure S6 and Fig 3d the Gaussians do not approach zero for high areas. I don't understand this. The integrated distribution must be unity. Here it is infinite.

3) The role of the chirality of the helicenes is totally unclear to the reader unless one reads the paragraph on page S11. This point should be mentioned early on in the paper.

Reviewer #2 (Remarks to the Author):

In this work the authors present a wet deposition protocol for manipulating the growth kinetics of

ultrathin Au films on the Au(111) surface which leads essentially to epitaxial growth with control over the shape and size of the film. Central to their approach is the usage of a mixture of solution-born nanometer-size gold clusters as the source of gold adatoms and helicenes containing thioether moieties which facilitate this atomic layer deposition at the liquid/metal interface. When the thiol-protected 38-atom Au cluster was dissolved in 1,2,4- trichlorobenzene and deposited on the Au(111), the STM images pointed to separation of the Au atoms from their organic ligands, leading to formation of Au islands but without any control over their shapes or lateral dimensions. On the other hand, when the (P)-1 enantiomer of diazadithia helicene is added to the thiol-protected 38-atom Au cluster solution in 1, 2, 4- trichlorobenzene, the deposition of the resulting mixture displays striking impact on the morphology of the Au films. The helicenes lead to the formation of triangular islands. Furthermore, the broad distribution of island sizes converts to a narrow one when the ratio of the helicene to the Au cluster is 2 or greater, as deduced from comparisons of observed experimental STM images with their computational counterparts. Another key observation is that higher concentration of the gold cluster leads to multiple-layer islands before a full atomic layer coverage is reached, pointing to the role of kinks. Further detailed analysis of STM images, surface structural parameters, bond-lengths and related DFT calculations of system energetics leads the authors to conclude that kink generation by the helicenes is responsible for these striking observations.

This is a nice piece of work, with novel results which would be of interest to researchers in the field. The authors have paid attention to a number of details in their analysis of this multiple component system, which help them make a compelling case. The authors would like to suggest that their proposed wet deposition technique should provide a recipe for manipulating the shape and size of ultrathin films in general. It would be good to establish this to be the case, but that is not yet conclusive from the set of observations made by the authors. Relative strengths of interactions of the different molecules with the surface and between each other also help determine growth kinetics and dynamics. More work needs to be done following the guidelines proposed by the authors before a general overarching principle can be deduced.

My recommendation is that the paper be accepted for publication if the authors are willing to restrict their conclusions to the system at hand. These are already interesting results which will motivate others to follow suite. There are a couple of places where a bit more clarity in the text would be helpful. For example, on page 5 isn't Au₃₈(SCH₂CH₂Ph)₂₄ the same as the racemic cluster, (rac)-Au₃₈? Why not make it clear? Similarly, on page 14 the authors write "Such a small energy difference is likely the reason that all triangles have the same orientation." Perhaps they would like to say "This energy difference, albeit small, is likely the reason that all triangles have the same orientation."

Reviewer #3 (Remarks to the Author):

This paper presents a significant new strategy in growing metallic nanostructures at the solution/solid interface. Wet deposition of gold cluster precursors is used to fabricate gold nanoislands on the surface. Furthermore, the authors demonstrate that helicenes in the same solution can adsorb on the nanoislands and steer their growth. This work is novel in presenting a solution based growth mode. The experimental work is well done and presents convincing images of island growth and control of that growth using helicene adsorption. Faceting is demonstrated and DFT calculations help to interpret adsorption strengths, etc. to rationalize the shape control. This paper will be of high interest to the readership of Nature Communications. It will demonstrate a new growth method and also important considerations in how to steer and control nanoisland growth. I recommend publication after addressing the following points.

The image sequence in Figures 4d, 4e, 4f is rather unconvincing since there seems to be concurrent changes in image quality that make it difficult to interpret whether the surface structure is actually changing or just the image quality / tip state. A more convincing argument for the presence of the

helicenes is shown in Figure S11.

Note also that the "kinks" indicated by arrow b in Figure 4f cannot be clearly distinguished from the helicene dots indicated by the other arrows, so this is not a strong argument to rationalize the kinks that were found in DFT.

It is not clear why helicenes would not be present and imaged in the interior of the islands?

In the last sentence of page 10, I do not think the authors mean "directional," as the islands do not have a specific direction. Perhaps "selective?"

Abstract: "born" or "borne?"

Revise caption to Figure 1d: the model is not "superimposed."

Reviewers' comments:

Reviewer #1 (Remarks to the Author):

The manuscript of Cao et al. reports on the growth of Au islands at the liquid-metal interface. The main result is the size distribution and shape of the islands grown in presence/absence of a molecular additive. The molecular additive is chiral, but its chirality is found to have no visible effect on the shape/faceting of the islands. A dependence on the helicity of the additive would have been a very interesting result. In the present state of the manuscript one wonders why such complex molecules have been chosen, it appears that many other additives would also yield the same effect.

The effect is interesting, but the paper raises a number of questions which should be addressed (see below). In the present state of the manuscript I'm not convinced if the size distribution is really related to the epitaxy. Here, the stabilization of smaller islands and 3D mound formation (at high concentrations) appears to be related to the stabilization of certain edges/facets (see 1). In that sense the work is similar to the cited literature on 'faceting'. On the other hand the deposition via Au clusters appears to be novel.

Author reply:

We thank the reviewer for his/her evaluation of our work.

The chirality of helicenes indeed does not lead to the formation of an asymmetrical surface morphology. We identify two possible reasons: 1) the pedant thioether groups rather than the helical aromatic backbone are acting on the atomic layer growth; 2) the helicenes at the step edges are side-by-side oriented along the symmetry directions of the surface. Therefore, no chiral bias is imparted to the island shape.

In vacuum-based metal epitaxy, gases (O and CO), Sb, Pb, and In have been applied as surfactant to exert control over thin layer growth. Such additives can interact sufficiently with the metal, float on top of the growing layer and be easily removed. At the liquid/metal interface, the diffusion of organic molecules can be significantly promoted by the fluids, therefore there are indeed a lot of organic molecules available. Our selection of helicenes as an additive is based on the extensive knowledge of the adsorption and organization of helicenes at metal surfaces (ref: Acc. Chem. Res. 2016, 49, 1182–1190). In addition, the characteristic bright feature of helicenes in STM images facilitates in unraveling the way they interact with the metal.

We noticed that the schematic illustration and the description of our methods are in part oversimplified. We have made a few changes to the manuscript and added the following two figures to show how atomic layer deposition is achieved in a drop of solution at room temperature.

a, Schematic illustration of the STM measurement at the liquid/Au(111) interface. All our STM measurements were carried out at room temperature in the presence of solvent (1,2,4-Trichlorobenzene), of which the boiling point amounts to 214 °C. The evaporation loss of solvent in hours of STM measurements can be neglected. Helicenes and Au clusters were transferred to the Au(111) surface via dropcasting their solutions in 1,2,4-Trichlorobenzene. **b, Growth of Au nanoislands via a wet deposition protocol.** The thiolate-protected Au cluster is a stable compound in organic solvent, but disassembles spontaneously at the liquid/Au(111) interface, due to the detachment of thiolate ligands at the solution/Au(111) interface (Figure b). No heating was applied.

1) Is this effect reversible/dynamic, i.e. does the size distribution change if the helicenes are added after the islands have been grown? This experiment will tell if one can consider the growth here as nucleation and growth or if it's an equilibrium process depending on the concentration of helicenes. Figure 6 hints towards an equilibrium process.

Author reply:

Actually, by the time the islands have grown, majority of the helicenes are in the solution phase while only a very small portion of helicenes are present on the surface, and they mainly appear at the island edges.

Take a 10 μL solution containing 25 μM helicene and 25 μM Au cluster for example, the helicenes in solution is theoretically able to cover the surface ($1 \times 1 \text{ cm}^2$) with up to 1.5 molecules per nm^2 (see Fig. S1 for a large scale image of the superstructures of helicenes). The experimentally observed density of helicenes on the surface is remarkably lower though. For instance, in Fig. 4b, less than 100 helicenes can be identified in a $40 \times 40 \text{ nm}^2$ image, which is only 4% of what is theoretically possible.

While the surface is covered with islands of different size, we usually found that changes to the islands occur at the steps (Fig. 3d-3f, now Fig. 4d-4f, main text) and to small clusters (Fig.6, now Fig. 7, main text and the image below).

Figure: The size of these images is $170 \times 170 \text{ nm}^2$. The blue arrows indicate the changes in 12 minutes.

The reversible effect, as presented in Fig. 6 (now Fig. 7), appears only on small clusters, which may otherwise grow or decay. This process is not obvious on the vast majority of islands having relatively larger sizes.

While it is likely that helicenes intervene both in the growth and decay of islands, the former process slows down when there are limited Au atoms/clusters in the solution phase. The latter process, on the other hand, should easily occur at sites with highly under-coordinated surface Au atoms. Note though that there is a significant barrier for helicenes to pull out Au atoms that are already in an “equilibrium” or in a stable state, especially considering that the mobility of top layer Au atoms of Au(111) surface is lower compared to other facets (110, 100, etc.) (ref: *Surf. Sci.* **253**, 334-344 (1991)).

In other words, the dynamic adsorption/desorption of helicenes is likely able to intervene in island growth at the initial phase of atomic layer deposition when there are plenty of Au atoms/clusters available in the solution and on the surface, but have no pronounced impact on the size distribution after stable islands are formed.

We agree with the reviewer (2nd comment) that the helicenes stabilize certain cluster sizes and allows for a higher nucleation density, leading to the formation of smaller islands. We have revised the discussion in the main manuscript.

2) If the growth is not dynamic, in vacuum based epitaxy the impingement rate of atoms to the surface vs. the diffusion at the surface have been considered. [1] In this picture, the ‘nucleation density’ is an important quantity. It could be that the presence of helicenes stabilizes certain cluster sizes and this allows for a higher nucleation density. This can be analyzed from the existing data.

Author reply:

We thank the reviewer for his/her insightful comments. Indeed there is an obvious correlation between the concentration of helicene and the density/size of islands, as displayed in the following figure. This figure accompanied with a brief discussion is now included in the revised manuscript.

3) Related to 2) also the dependence on the metal source (Au cluster concentration) which should probably regulate the rate at which the Au arrives is not discussed. Or is the rate at which Au arrives at the surface defined by the evaporation rate of the solvent? The methodical part of the paper is not very clear in my opinion.

Author reply:

We believe there is a misunderstanding of the experimental details. We have revised the schematic illustration (now Figure 1). The evaporation of solvent (b.p. 214 °C) is very slow at RT, and the loss of solvent that occurs during the STM experiment can be neglected in the course of an STM experiment.

At the liquid/solid interface, the number of molecules (n) arriving on the surface per unit time (t) can be expressed by

$$\frac{-dn}{A dt} = \frac{D(C_l - C_s)}{\delta}$$

Where A is the surface area, D is the diffusion constant of the molecule, which can be determined by the Stokes–Einstein equation. C_l and C_s refer to the concentrations of these molecules in the

liquid phase and at the surface, respectively. δ is the effective thickness of diffusion layer (Nernst's diffusion layer). (ref: Reactions at the Liquid-Solid Interface, Vol 28, 1989, Elsevier)

Au can exist in the liquid phase as intact cluster or as Au atoms that are complexed with ligands. Upon depositing the mixture of Au clusters and helicenes on Au(111) surface, there are no intact Au clusters on the surface and most of the helicenes stay in the liquid phase, therefore for both compounds $C_s \approx 0$ and $C_l - C_s \approx C_l$. As such, the rate at which Au clusters arriving at the surface is roughly determined by their concentration in the solution. The concentration of helicene, on the other hand, can affect the assembly of islands by forming a complex with the Au atoms.

4) Can the effect of the helicenes be discussed in terms a “surfactant”? The presence of helicenes clearly modifies the step/surface energies.

[1] Harald Brune and Klaus Kern; Chapter 5 Heteroepitaxial metal growth: the effects of strain; The Chemical Physics of Solid Surfaces 8; 149 (1997).

Author reply: We totally agree with the reviewer that the helicenes act as a “surfactant”. We have discussed this in the revised manuscript.

Other comments:

1) The figures are not reader-friendly: e.g. in Fig. 3d the reader has to extract the meaning of the curves from the caption. It would be much easier to give this information in the legend. The other figures can be also improved by adding some text to the figures themselves.

Author reply:

We have modified the figure accordingly. The effect of helicene on island size and density now is provided, and we have added text to the figure to make it easier to read.

2) In Figure S6 and Fig 3d the Gaussians do not approach zero for high areas. I don't understand this. The integrated distribution must be unity. Here it is infinite.

Author reply: We thank the reviewer for his/her careful reading. We applied curve fitting to fit the statistics. The fitting function is

$$y = y_0 + \frac{A}{w\sqrt{\pi/2}} e^{-2\frac{(x-x_c)^2}{w^2}}$$

The fitting curve is therefore different to the normal distribution curve, of which the integrated distribution is unity.

We have now provided the normal distribution of island areas at different helicene concentrations.

3) The role of the chirality of the helicenes is totally unclear to the reader unless one reads the paragraph on page S11. This point should be mentioned early on in the paper.

Author reply: we have now mentioned in the main manuscript that the chirality of the helicenes has no significant effect on the growth preference of islands.

Reviewer #2 (Remarks to the Author):

In this work the authors present a wet deposition protocol for manipulating the growth kinetics of ultrathin Au films on the Au(111) surface which leads essentially to epitaxial growth with control over the shape and size of the film. Central to their approach is the usage of a mixture of solution-born nanometer-size gold clusters as the source of gold adatoms and helicenes containing thioether moieties which facilitate this atomic layer deposition at the liquid/metal interface. When the thiol-protected 38-atom Au cluster was dissolved in 1,2,4- trichlorobenzene and deposited on the Au(111), the STM images pointed to separation of the Au atoms from their organic ligands, leading to formation of Au islands but without any control over their shapes or lateral dimensions. On the other hand, when the (P)-1 enantiomer of diazadithia helicene is added to the thiol-protected 38-atom Au cluster solution in 1, 2, 4- trichlorobenzene, the deposition of the resulting mixture displays striking impact on the morphology of the Au films. The helicenes lead to the formation of triangular islands. Furthermore, the broad distribution of island sizes converts to a narrow one when the ratio of the helicene to the Au cluster is 2 or greater, as deduced from comparisons of observed experimental STM images with their computational counterparts. Another key observation is that higher concentration of the gold cluster leads to multiple-layer islands before a full atomic layer coverage is reached, pointing to the role of kinks. Further detailed analysis of STM images, surface structural parameters, bond-lengths and related DFT calculations of system energetics leads the authors to conclude that kink generation by the helicenes is responsible for these striking observations.

This is a nice piece of work, with novel results which would be of interest to researchers in the field. The authors have paid attention to a number of details in their analysis of this multiple component system, which help them make a compelling case. The authors would like to suggest that their proposed wet deposition technique should provide a recipe for manipulating the shape and size of ultrathin films in general. It would be good to establish this to be the case, but that is not yet conclusive from the set of observations made by the authors. Relative strengths of interactions of the different molecules with the surface and between each other also help determine growth kinetics and dynamics. More work needs to be done following the guidelines proposed by the authors before a general overarching principle can be deduced.

My recommendation is that the paper be accepted for publication if the authors are willing to restrict their conclusions to the system at hand. These are already interesting results which will motivate others to follow suite. There are a couple of places where a bit more clarity in the text would be helpful. For example, on page 5 isn't Au₃₈(SCH₂CH₂Ph)₂₄ the same as the racemic

cluster, (rac)-Au₃₈? Why not make it clear? Similarly, on page 14 the authors write “Such a small energy difference is likely the reason that all triangles have the same orientation.” Perhaps they would like to say “This energy difference, albeit small, is likely the reason that all triangles have the same orientation.”

Author reply:

We thank the reviewer for his/her positive evaluation of our manuscript, the kind suggestions, and the recommendation of accepting this work.

We have modified the manuscript to restrict our conclusions to the current system.

We have clarified the relationship between “Au₃₈(SCH₂CH₂Ph)₂₄” and (rac)-Au₃₈, and revised the sentence, according to the reviewer’s suggestion.

Reviewer #3 (Remarks to the Author):

This paper presents a significant new strategy in growing metallic nanostructures at the solution/solid interface. Wet deposition of gold cluster precursors is used to fabricate gold nanoislands on the surface. Furthermore, the authors demonstrate that helicenes in the same solution can adsorb on the nanoislands and steer their growth. This work is novel in presenting a solution based growth mode. The experimental work is well done and presents convincing images of island growth and control of that growth using helicene adsorption. Faceting is demonstrated and DFT calculations help to interpret adsorption strengths, etc. to rationalize the shape control. This paper will be of high interest to the readership of Nature Communications. It will demonstrate a new growth method and also important considerations in how to steer and control nanoisland growth. I recommend publication after addressing the following points.

Author reply:

We thank the reviewer for his/her positive evaluation of our manuscript and the recommendation of accepting this work.

The image sequence in Figures 4d, 4e, 4f is rather unconvincing since there seems to be / changes in image quality that make it difficult to interpret whether the surface structure is actually changing or just the image quality / tip state. A more convincing argument for the presence of the helicenes is shown in Figure S11.

Author reply:

We have now provided in Supplementary Information the larger scale images of Figures 4d, 4e, 4f, in which the surrounding surface structures can be used as reference for the tip state. Similar changes as displayed in Fig. 4d-4f (now Fig. 5d-5f) can be identified from the neighboring islands. We have also relocated the STM image in Figure S11 to the main manuscript to show that the bright dots on top of islands are indeed helicenes.

Note also that the “kinks” indicated by arrow b in Figure 4f cannot be clearly distinguished from the helicene dots indicated by the other arrows, so this is not a strong argument to rationalize the kinks that were found in DFT.

Author reply:

Figure 4f shows the formation of two helicene rows along a step of an island, one on the terrace plane of the substrate and one at the step edge of the island. The bright feature of the kink is ambiguous, therefore the kinks that are surrounded by two helicene rows were not involved in DFT simulations, and only a tentative model for the surface structure was presented.

DFT modelling on the kink formation was based on the high-resolution STM image in Figure 4g, where the adsorption situation of helicenes can be roughly identified from their characteristic bright feature.

It is not clear why helicenes would not be present and imaged in the interior of the islands?

Author reply:

According to our DFT calculation, the attachment of Au adatoms to a step that is decorated with helicenes would weaken the chemical adsorption of helicenes. That is to say, the adsorption of helicene in the interior of an island should be weaker than that at the step, therewith the mobility of helicenes in the interior is higher. It is now mentioned in the main text.

In the last sentence of page 10, I do not think the authors mean “directional,” as the islands do not have a specific direction. Perhaps “selective?”

Author reply: we thank the reviewer for his/her careful readings, we have replaced “directional” with “selective”.

Abstract: “born” or “borne?”

Author reply:

We have change it to “solution-borne”.

Revise caption to Figure 1d: the model is not “superimposed.”

Author reply: the caption has been changed.

Reviewers' comments:

Reviewer #1 (Remarks to the Author):

As discussed before and mentioned by the other reviewers, the paper present a novel way to control the size distribution in solution based epitaxy. This result is exciting. The revised version of the MS has significantly improved both in the interpretation/explanation of the results as well as in their presentation.

I still don't understand the size distribution, its fits and the extracted average sizes. Since this is the main result of this paper, a more substantiated analysis is in my opinion very important. Provided that the issues with the analysis are resolved (!), I can recommend publication.

Concerning the size distribution (Figure 4):

1) The new fits using a normal distribution certainly make more sense than the used function including an offset y_0 . However, the integral of the distributions are again not unity, because there are no negative island sizes.

2) The data (Fig. 4e-h) however appears to follow a $1/x$ relationship. In all cases the smaller islands are more frequent than bigger ones, it's just that with higher concentrations the decay is flatter.

3) Because of b) I don't understand why the maxima of the fits are not very close to zero. E.g. in absence of helicenes, in the previous version the maxima of the fit was around 100 nm^2 (Fig. S6, black), but now it's around 200 nm^2 .

4) The fit function from where the size is extracted should reproduce the data reasonably well. Alternatively, the average/median size could be reported without any fit. The median is probably better, because it's robust against rarely detected large islands.

5) I cannot find the definition of the fit function in the manuscript.

6) Figure 4i: I don't understand how the island size is related to the data. Without helicenes, I read around 300 nm^2 for the size of an average island from Fig. 4e. Fig. 4i shows 700 nm^2 . Likewise for 50 uM helicenes (Fig. 4h) I read around 50 nm^2 but Fig. 4i shows 250 nm^2 . In this case 50 nm^2 is far out of the error bars. The frequencies (4e-h) for the average sizes reported in Fig. 4i are all virtually zero.

Reviewer #3 (Remarks to the Author):

In this revision, the authors have adequately and completely addressed the concerns raised by each of the referees. In considering the revised manuscript and the revisions to the SI, I think that the manuscript now clearly merits publication in Nature Communications and will make an important contribution to the literature. This work will be of broad interest to the readership of Nature Comms and presents an important conceptual advance in our understanding of growth at surfaces. I recommend publication in its current form.

Reviewers' comments:

Reviewer #1 (Remarks to the Author):

As discussed before and mentioned by the other reviewers, the paper present a novel way to control the size distribution in solution based epitaxy. This result is exciting. The revised version of the MS has significantly improved both in the interpretation/explanation of the results as well as in their presentation.

I still don't understand the size distribution, its fits and the extracted average sizes. Since this is the main result of this paper, a more substantiated analysis is in my opinion very important. Provided that the issues with the analysis are resolved (!), I can recommend publication.

Author reply: we thank the reviewer again for his/her careful reading of our manuscript and all the valuable comments.

Concerning the size distribution (Figure 4):

- 1) The new fits using a normal distribution certainly make more sense than the used function including an offset y_0 . However, the integral of the distributions are again not unity, because there are no negative island sizes.

Author reply: All the statistics in this study are not normally distributed, therefore the integral of the distributions are not unity.

- 2) The data (Fig. 4e-h) however appears to follow a $1/x$ relationship. In all cases the smaller islands are more frequent than bigger ones, it's just that with higher concentrations the decay is flatter.

Author reply: Yes, indeed the statistics implies a nearly inverse relationship between the island size and the helicene concentration. The islands are mostly irregular in the absence of helicenes and the island size spans a very wide range, which is significantly narrowed to smaller values while using helicenes as an additive.

- 3) Because of b) I don't understand why the maxima of the fits are not very close to zero. E.g. in absence of helicenes, in the previous version the maxima of the fit was around 100 nm^2 (Fig. S6, black), but now it's around 200 nm^2 .

Author reply: The discrepancy is likely caused by the different functions we applied for the fits. The curve fitting function we previously applied included an additional y_0 offset, and the maximum is more close to the sizes having the highest frequencies. A comparison of the two methods is displayed in the following:

- 4) The fit function from where the size is extracted should reproduce the data reasonably well. Alternatively, the average/median size could be reported without any fit. The median is probably better, because it's robust against rarely detected large islands.

Author reply: We agree that the fitting functions we used might be confusing and cannot adequately describe the effect of helicene on island size distribution. Therefore, **we have removed all the fits in Figure 4 and use median/average sizes instead**, as recommended by the referee

- 5) I cannot find the definition of the fit function in the manuscript.

Author reply: The nonlinear curve fitting we previously applied is to a certain extent similar to the regression analysis function, and was intended to construct a curve that has the best fit to a series of data points. Following your recommendation, we have removed these fits.

$$y = y_0 + \frac{A}{w\sqrt{\pi/2}} e^{-2\frac{(x-x_c)^2}{w^2}}$$

- 6) Figure 4i: I don't understand how the island size is related to the data. Without helicenes, I read around 300 nm² for the size of an average island from Fig. 4e. Fig. 4i shows 700 nm². Likewise for 50 uM helicenes (Fig. 4h) I read around 50 nm² but Fig. 4i shows 250 nm². In this case 50 nm² is far out of the error bars. The frequencies (4e-h) for the average sizes reported in Fig. 4i are all virtually zero.

Author reply: We sincerely apologize for a mistake we made: these mean values should have been multiplied by the coverage of the islands, which is typically around 20%. Figure 4i has now been corrected. This correction does not affect any of the discussions or conclusions as presented in the manuscript. We really appreciate the reviewer for pointing this out.

We have now provided details of the statistics in the supplementary information (Table S3-S7), including all the necessary information concerning the size distribution, and the data that have been used for the analysis.

Reviewer #3 (Remarks to the Author):

In this revision, the authors have adequately and completely addressed the concerns raised by each of the referees. In considering the revised manuscript and the revisions to the SI, I think that the manuscript now clearly merits publication in Nature Communications and will make an important contribution to the literature. This work will be of broad interest to the readership of Nature Comms and presents an important conceptual advance in our understanding of growth at surfaces. I recommend publication in its current form.

Author reply: we thank the reviewer for his/her evaluation of our manuscript and the recommendation of accepting our manuscript.

REVIEWERS' COMMENTS:

Reviewer #1 (Remarks to the Author):

The only remaining issue in the manuscript was the concerning the island sizes (Figure 4). The authors chose to remove the fits and report the median values, as suggested. Most importantly the island sizes (Figure 4i) correspond to the reported values. The values are also reported in the SI.

The authors correctly point out that the wrong values did not affect the conclusions of the paper.

PS: I still think that any model (i.e. fit) of a probability distribution (like here) should have an integral of unity. This is however irrelevant since the fits have been removed.

I can now recommend publication of these very nice results! I apologize for taking more than one week for this review.

REVIEWERS' COMMENTS:

Reviewer #1 (Remarks to the Author):

The only remaining issue in the manuscript was the concerning the island sizes (Figure 4). The authors chose to remove the fits and report the median values, as suggested. Most importantly the island sizes (Figure 4i) correspond to the reported values. The values are also reported in the SI.

The authors correctly point out that the wrong values did not affect the conclusions of the paper.

PS: I still think that any model (i.e. fit) of a probability distribution (like here) should have an integral of unity. This is however irrelevant since the fits have been removed.

I can now recommend publication of these very nice results! I apologize for taking more than one week for this review.

Author reply: we thank the reviewer again for carefully reading of our manuscript and providing valuable advices.